# Batch Bayesian Optimization on Permutations using the Acquisition Weighted Kernel

**Changyong Oh**
QUvA lab, IvI
University of Amsterdam
changyong.oh0224@gmail.com

**Roberto Bondesan**
Qualcomm AI Research*
rbondesa@qti.qualcomm.com

**Efstratios Gavves**
QUvA lab, IvI
University of Amsterdam
egavves@uva.nl

**Max Welling**
QUvA lab, IvI
University of Amsterdam
m.welling@uva.nl

## Abstract

In this work we propose a batch Bayesian optimization method for combinatorial problems on permutations, which is well suited for expensive-to-evaluate objectives. We first introduce LAW, an efficient batch acquisition method based on determinantal point processes using the acquisition weighted kernel. Relying on multiple parallel evaluations, LAW enables accelerated search on combinatorial spaces. We then apply the framework to permutation problems, which have so far received little attention in the Bayesian Optimization literature, despite their practical importance. We call this method LAW2ORDER. On the theoretical front, we prove that LAW2ORDER has vanishing simple regret by showing that the batch cumulative regret is sublinear. Empirically, we assess the method on several standard combinatorial problems involving permutations such as quadratic assignment, flowshop scheduling and the traveling salesman, as well as on a structure learning task.

## 1 Introduction

From the celebrated traveling salesman problem[GP06] to flowshop and jobshop scheduling problems[GJS76], permutations are ubiquitous representations in combinatorial optimization. Such combinatorial problems on permutations arise in highly impactful application areas. For example, in chip design, permutations specify relative placements of memories and logical gates on a chip[AMS08]. As another example, in 3D printing, scheduling is an important factor to determine the production time[CHHV18, GSE+19, SFZ+20]. In both cases, as well as in many others, evaluating the cost associated to a given permutation is expensive.

In situations where the evaluation is expensive, Bayesian optimization (BO) has shown good performance in many problems[SLA12, SRS+15]. Recently, BO on combinatorial spaces has made significant progress for categorical variables [BP18, DBDF20, OTGW19, SRDM20, DSR+20]. However, BO on permutations is yet under-explored with a few exceptions [ZSBB14, ZYLW19, BBG+20].

In this work we present a framework to deal with BO on permutations where the evaluation of the objective is expensive. We extend batch Bayesian optimization, which allows one to speed up the optimization by acquiring a batch of multiple points and evaluating the batch in parallel[AFF10, GDHL16], to the case of permutations. Then, motivated by the observation that both the diversity of

---

*Qualcomm AI Research is an initiative of Qualcomm Technologies, Inc.

the points in the batches and the informativeness of the individual points in the batch improve the performance[GPL19], we propose a new batch acquisition method which is applicable to the search space of permutations and takes into account both the diversity of the batch and the informativeness of each point. This method is based on determinantal point processes (DPPs), which have been widely used to model sets of diverse points[KT12], and can be conveniently incorporated into the Gaussian Process framework since DPPs are specified by a kernel. To overcome the lack of informativeness of DDPs[KDK16] (more specifically, the selection of points in batches relies solely on the predictive variance of the surrogate model), we enhance DPPs by using a kernel weighted by acquisition values. Therefore, we propose a new batch acquisition method using the so-called DPP **L**-ensemble[BR05, KT12] augmented with the **A**cquisition **W**eight, dubbed LAW. The whole procedure to find the optimal ordering (permutation) through LAW is thus dubbed LAW2ORDER. We compare LAW2ORDER and other competitors, firstly, on three combinatorial optimization benchmarks on permutations such as quadratic assignment problem, the flowshop scheduling problem, and the traveling salesman problem. We also make comparisons on the structure learning problem. In the structure learing problem, LAW2ORDER performs the best and the performance gap is more significant for larger permutation spaces. Moreover, LAW2ORDER still outperforms significantly genetic algorithms which use twice as many evaluations.

## 2 Preliminaries

In this section, we briefly discuss some prerequisites for our proposed method and introduce notation. Below we will denote a function $f$ with one input as $f(\cdot)$, and function $K$ with two inputs as $K(\cdot, \cdot)$. For $B \in \mathbb{N}$, $[B] = \{1, \cdots, B\}$ while for a set $\mathcal{X}$, $|\mathcal{X}|$ is the number of elements in $\mathcal{X}$.

### 2.1 Batch Bayesian Optimization

Bayesian Optimization (BO) aims at finding the global optimum of a black-box function $f$ over a search space $\mathcal{X}$, namely, $\mathbf{x}_{opt} = \operatorname{argmin}_{\mathbf{x} \in \mathcal{X}} f(\mathbf{x})$. Two main components are the probabilistic modeling of the objective $f(\mathbf{x})$ and the acquisition of new points to evaluate. Probabilistic modeling is performed by the surrogate model. At the $t$-th round, the surrogate model attempts to approximate $f(\mathbf{x})$ based on the evaluation data $\mathcal{D}_{t-1}$, producing the predictive mean $\mu_{t-1}(\mathbf{x}) = \mu(\mathbf{x} \mid \mathcal{D}_{t-1})$ and the predictive variance $\sigma_{t-1}^2(\mathbf{x}) = \sigma^2(\mathbf{x} \mid \mathcal{D}_{t-1})$. In the acquisition of a new point, the acquisition function $a_t(\mathbf{x}) = a_{seq}(\mathbf{x} \mid \mu_{t-1}(\cdot), \sigma_{t-1}^2(\cdot))$ is specified, which is based on the predictive mean $\mu_{t-1}(\cdot)$ and the predictive variance $\sigma_{t-1}^2(\cdot)$ to score how informative points are for the optimization. Next, the point that maximizes the acquisition function is obtained, $\mathbf{x}_t = \operatorname{argmax}_{\mathbf{x}} a_t(\mathbf{x})$, and the objective evaluated, $y_t = f(\mathbf{x}_t)$. Then, the new evaluation point is appended to the old dataset, $\mathcal{D}_t = \mathcal{D}_{t-1} \cup \{(\mathbf{x}_t, y_t)\}$ and the process repeats by fitting the surrogate model with $\mathcal{D}_t$. The process continues until the evaluation budget is depleted. To contrast with the proposed method, we call this basic BO as sequential BO. For a more extensive overview of Bayesian optimization, please refer to [SSW$^+$15, Fra18].

With more computational resources, such as more GPUs and CPUs, we can speed up Bayesian optimization by allowing multiple evaluations in parallel. For this, we acquire a batch of multiple points, a method known as Batch Bayesian Optimization (BBO)[AFF10, GDHL16]. In BBO, we need an acquisition function $a_{batch}$ scoring the quality of batches of $B$ points $\{\mathbf{x}_b\}_{b \in [B]}$ instead of individual points. At time $t$, a batch of $B$ points is acquired $\{\mathbf{x}_{t,b}\}_{b \in [B]} = \operatorname{argmax}_{\{\mathbf{x}_b\}_{b \in [B]}} a_t(\{\mathbf{x}_b\}_{b \in [B]})$, where $a_t(\{\mathbf{x}_b\}_{b \in [B]}) = a_{batch}(\{\mathbf{x}_b\}_{b \in [B]} \mid \mu_{t-1}(\cdot), \sigma_{t-1}^2(\cdot))$. Then the points in the acquired batch are evaluated in parallel and the evaluation data is updated by $\mathcal{D}_t = \mathcal{D}_{t-1} \cup \{(\mathbf{x}_{t,b}, y_{t,b})\}_{b \in [B]}$.

### 2.2 Determinantal Point Processes

Determinantal point processes (DPPs) are stochastic point processes well-suited to model sets of diverse points[KT12]. Let us assume that we want to sample a set of diverse points from a finite set $\mathcal{X}$. One way to define DPP is to use the so-called L-ensemble[KT12]. For a given kernel $L(\cdot, \cdot)$ on $\mathcal{X}$, the L-ensemble is defined as the random point process with density $P_L^{DPP}(X) = \frac{\det([L(\mathbf{x}, \mathbf{y})]_{\mathbf{x}, \mathbf{y} \in X})}{\det(I+L)}$ where $X \subset \mathcal{X}$ and $[L(\mathbf{x}, \mathbf{y})]_{\mathbf{x}, \mathbf{y} \in X}$ is a submatrix of $L$ restricted to $X$[BR05].

For a batch of just two points, $X = \{\mathbf{x}, \mathbf{y}\}$ it is easy to observe that DPP encourages diversity — $P_L^{DPP}(\{\mathbf{x}, \mathbf{y}\}) \propto L(\mathbf{x}, \mathbf{x})L(\mathbf{y}, \mathbf{y}) - L(\mathbf{x}, \mathbf{y})^2$. Indeed, for more similar points the value of $L(\mathbf{x}, \mathbf{y})$ is higher, resulting in a lower density. In DPPs there is no cardinality constraint on $X$. We,

therefore, define $k$-DPP, which is a DPP with the restriction that sampled sets have precisely $k$ points. Denoting the set of subsets of $\mathcal{X}$ with $k$ points by $\mathcal{X}_k$, the $k$-DPP density is defined for $X \in \mathcal{X}_k$ by $P_L^{k\text{-}DPP}(X) = \frac{\det([L(\mathbf{x},\mathbf{y})]_{\mathbf{x},\mathbf{y} \in X})}{\sum_{X' \in \mathcal{X}_k} \det([L(\mathbf{x},\mathbf{y})]_{\mathbf{x},\mathbf{y} \in X'})}$. Therefore, $X^* = \operatorname{argmax}_{X \in \mathcal{X}_k} P_L^{k\text{-}DPP}(X)$ is the most diverse set of $k$ points with respect to the similarity encoded by the kernel $L(\cdot, \cdot)$.

In our algorithm, we use that log of k-DPP density is submodular[SKKS09, KT12] and can be greedily maximized with approximation guarantees[NWF78, Sak20] (See Supp.Subsec. F.1 for a brief discussion).

# 3   Method

Batch acquisition on a combinatorial space poses two difficulties. First, the batch acquisition objectives of existing batch Bayesian optimization are designed based on the properties and intuition applicable to continuous spaces[GDHL16, GPL19]. This may not always be suitable for discrete spaces. For instance, the method in [GDHL16] is defined by using Euclidean distance. Also, the difficulties of combinatorial optimization are exacerbated when optimizing a batch jointly. This is in stark contrast to the continuous case where gradient based optimization is easily extended to batch optimization of multiple points in parallel[WCLF20].

To cope with these challenges, we introduce a new batch acquisition method for Bayesian optimization, the maximization of the determinantal point process (DPP) density defined by an **L**-ensemble with **A**cquisition **W**eights, dubbed **LAW**. We describe LAW in Subsec. 3.1 and its regret analysis is provided in Subsec. 3.2 and 3.3.

## 3.1   Batch Acquisition using LAW

We start to define the main components of LAW.

**Definition 3.1** (Weight function). We call a function $w : \mathbb{R} \to \mathbb{R}$ a weight function if it is positive ($r \in \mathbb{R}, w(r) > 0$), increasing ($r_1 \leq r_2 \implies w(r_1) \leq w(r_2)$), and bounded below and above by a positive number ($w_- = \inf_{r \in \mathbb{R}} w(r) > 0$ and $w_+ = \sup_{r \in \mathbb{R}} w(r) < \infty$).

**Definition 3.2** (Posterior covariance function). Given a (prior) kernel $K(\mathbf{x}_1, \mathbf{x}_2)$, data $\mathcal{D}$ and noise variance $\sigma^2$, the posterior (predictive) covariance function $K_{post}(\mathbf{x}_1, \mathbf{x}_2 \,|\, \mathcal{D}, \sigma^2)$ is defined as $K(\mathbf{x}_1, \mathbf{x}_2) - K(\mathbf{x}_1, \mathcal{D})(K(\mathcal{D}, \mathcal{D}) + \sigma^2 I)^{-1} K(\mathcal{D}, \mathbf{x}_2)$.

Let us assume that we are running batch Bayesian optimization with Gaussian process surrogate model using the kernel $K(\cdot, \cdot)$ and the acquisition function $a(\cdot)$, and that we acquire a batch of $B$ points in each round. At the $t$-th round, we have the evaluation data $\mathcal{D}_{t-1}$, the posterior covariance function $K_t(\cdot, \cdot) = K_{post}(\cdot, \cdot | K, \mathcal{D}_{t-1}, \sigma_{obs}^2)$ and the acquisition function $a_t(\{\mathbf{x}_b\}_{b \in [B]}) = a(\{\mathbf{x}_b\}_{b \in [B]} | \mu_{t-1}(\cdot), \sigma_{t-1}^2(\cdot))$ where $\mu_{t-1}(\cdot)$ and $\sigma_{t-1}^2(\cdot)$ are the predicitve mean and the predictive variance conditioned on $\mathcal{D}_{t-1}$.

In the existing work on batch Bayesian optimization using DPP[KDK16], the posterior covariance function is used as the kernel defining DPP. Even though the use of DPP in[KDK16] encourages diversity among points in batches, it essentially chooses points of high predictive variance.

However, the predictive mean also provides valuable information in Bayesian optimization. It is the acquisition function which harmonizes the predictive mean and the predictive variance to quantify how useful each point is. Therefore, we propose a new batch acquisition method which actively uses the acquisition function while retaining the strength of DPP encouraging diversity in each batch.

We define the *Acquisition Weighted* kernel $L^{AW}$ as follows
$$L^{AW}(\mathbf{x}_1, \mathbf{x}_2) = w(a(\mathbf{x}_1)) \cdot L(\mathbf{x}_1, \mathbf{x}_2) \cdot w(a(\mathbf{x}_2)).$$
Here $w$ is a positive weight function. We call the kernel $L$ in $L^{AW}$ the *diversity gauge* of $L^{AW}$.

With the posterior covariance function as the diversity gauge $L = K_t$ and the acquisition function $a = a_t$, the acquisition weighted kernel becomes
$$L_t^{AW}(\mathbf{x}_1, \mathbf{x}_2) = w(a_t(\mathbf{x}_1)) \cdot K_t(\mathbf{x}_1, \mathbf{x}_2) \cdot w(a_t(\mathbf{x}_2)) \tag{1}$$
Due to the dependency of $L = K_t$ and $a_t$ to the round index $t$, we subscript $L^{AW}$ and $L$ with $t$.

We optimize the density of DPP using the acquisition weighted kernel to choose the points in the batch. Before introducing how this optimization is actually performed, we provide an interpretation by rewriting the numerator of the density of the $k$-DPP defined by $L_t^{AW}$ as

---

**Algorithm 1** Batch Acquisition by $LAW$

---
1: **Input: weight function $w(\cdot)$, diversity gauge $L(\cdot, \cdot)$, acquisition function $a(\cdot)$, batch size $B$**
2: **Output: batch of $B$ points $\{\mathbf{x}_{t,1}, \cdots, \mathbf{x}_{t,B}\}$**
3: $\mathbf{x}_{t,1} = \operatorname{argmax}_{\mathbf{x} \in \mathcal{X}} a(\mathbf{x}) = \operatorname{argmax}_{\mathbf{x} \in \mathcal{X}} w(a(\mathbf{x}))$
4: **for** $b = 2, \cdots, B$ **do**
5: $\mathbf{x}_{t,b} = \operatorname{argmax}_{\mathbf{x}} \log(L(\mathbf{x}, \mathbf{x} | \{\mathbf{x}_{t,i}\}_{i \in [b-1]})) w(a(\mathbf{x}))^2$
6: **end for**

---

$$\det([L_t^{AW}(\mathbf{x}_i, \mathbf{x}_j)]_{i,j \in [B]}) = \det([K_t(\mathbf{x}_i, \mathbf{x}_j)]_{i,j \in [B]}) \prod_{i=1}^{B} w(a_t(\mathbf{x}_i))^2 \qquad (2)$$

This shows that the maximization of eq. (2) can be obtained by increasing both $\det([L_t(\mathbf{x}_i, \mathbf{x}_j)]_{i,j \in [B]})$ and $\prod_{i=1}^{B} w(a_t(\mathbf{x}_i))^2$ in a balanced way. Increasing the determinant term and increasing the product term promote diversity and acquisition values, respectively.

Now, we provide details on how the optimization is implemented. In combinatorial spaces where the optimization of a single point is challenging, the joint optimization of multiple points is daunting. Thanks to the submodularity of the log of the determinant[KT12], the joint optimization of multiple points can be approximated by a sequence of single point optimizations with an approximation guarantee (See Supp.Subsec. F.1 for submodularity).

The first point is obtained as in sequential Bayesian optimization by optimizing an acquisition function (line 3 in Alg. 1). The rest of the $B - 1$ points are obtained by maximizing the $k$-DPP density defined by the acquisition weighted kernel, which we approximately perform with a greedy method (line 4 in Alg. 1). Having chosen $b - 1$ points $\{\mathbf{x}_{t,i}\}_{i \in [b-1]}$, the greedy maximization selects $\mathbf{x}_b$ as follows:

$$\mathbf{x}_b = \operatorname*{argmax}_{\mathbf{x} \in \mathcal{X}} \log \det([L_t^{AW}(\cdot, \cdot)]_{\{\mathbf{x}_i\}_{i \in [b-1]} \cup \{\mathbf{x}\}}) = \operatorname*{argmax}_{\mathbf{x} \in \mathcal{X}} \log(L_t(\mathbf{x}, \mathbf{x} | \{\mathbf{x}_{t,i}\}_{i \in [b-1]}) \cdot w(a_t(\mathbf{x})^2))$$

where $L_t(\mathbf{x}, \mathbf{x} | \{\mathbf{x}_{t,i}\}_{i \in [b-1]})$ is the posterior variance of the kernel $L_t$ conditioned on $\{\mathbf{x}_{t,i}\}_{i \in [b-1]}$.

### 3.2 Regret Analysis

In this subsection, we provide a theoretical analysis on the performance of LAW with two acquisition functions, GP-UBC[SKKS09] and EST[WZJ16].

We begin with definitions needed in the analysis.

**Definition 3.3.** In the minimization of $f$ using batch acquisition, where $x^* = \operatorname{argmin}_{\mathbf{x}} f(\mathbf{x})$, $r_{t,b} = f(\mathbf{x}_{t,b}) - f(\mathbf{x}^*)$ is called instantaneous regret and $r_t^{(B)} = \min_{b \in [B]} r_{t,b} = \min_{b \in [B]} (f(\mathbf{x}_{t,b}) - f(\mathbf{x}^*))$ is called batch instantaneous regret. Simple regret is defined as the minimum of batch instantaneous regrets $R_T^{(B)}$.

$$S_T^{(B)} = \min_{t=1, \cdots, T} r_t^{(B)} = \min_{t=1, \cdots, T} \min_{b \in [B]} r_{t,b} \qquad (3)$$

Batch cumulative regret $R_T^{(B)}$ is defined as the sum of batch instantaneous regrets

$$R_T^{(B)} = \sum_{t=1}^{T} r_t^{(B)} = \sum_{t=1}^{T} \min_{b \in [B]} r_{t,b}. \qquad (4)$$

*Remark* 3.4. Note that $S_T^{(B)} \leq \frac{1}{T} R_T^{(B)}$. Vanishing simple regret is proved by showing $\frac{1}{T} R_T^{(B)} \to 0$.

**Definition 3.5.** For Gaussian processes with the kernel $K$ and the variance of observation noise $\sigma^2$, the maximum information gain $\gamma_T$ is defined as

$$\gamma_T = \gamma(T; \mathcal{X}, K, \sigma^2) = \max_{X \subset \mathcal{X}, |X| = T} \frac{1}{2} \log \det(I + \sigma^{-2} K(X, X)). \qquad (5)$$

For UCB and EST, we have the following regret bound.

**Theorem 3.6.** *Assume a kernel such that $K(\cdot, \cdot) \leq 1$, $|\mathcal{X}| < \infty$ and $f : \mathcal{X} \to \mathbb{R}$ is sampled from $\mathcal{GP}(\mathbf{0}, K)$. In each round $t \in [T]$ of batch Bayesian optimization, LAW acquires a batch using the evaluation data $\mathcal{D}_{t-1}$, the diversity measure $L_t(\cdot, \cdot) = K(\cdot, \cdot | \mathcal{D}_{t-1})$, an acquisition function $a_t(\cdot)$ and a weight function $w(\cdot)$ (Def. 3.1).*
*Let $C_1 = \frac{36}{\log(1+\sigma^{-2})}$ where $\sigma^2$ is the variance of the observation noise and $\delta \in (0, 1)$.*

At round $t$, define $\beta_{t,1}^{(B)UCB} = 2\log\left(\frac{|\mathcal{X}|\pi^2((t-1)B+1)^2}{6\delta}\right)$ and $\nu_t = \min_{\mathbf{x}}\left(\frac{\mu_{t-1}(\mathbf{x})-\hat{m}_t}{\sigma_{t-1,1}(\mathbf{x})}\right)$ where $\hat{m}_t$ is

the estimate of the optimum [WZJ16], $\zeta_t = \left(2\log\left(\frac{\pi_t^2}{2\delta}\right)\right)^{1/2}$, $\pi_t > 0$ such that $\sum_{t=1}^{\infty}\pi_t^{-1} \leq 1$.
Then batch cumulative regret satisfies the following bound

$$P\left(\left\{\frac{R_T^{(B)}}{T} \leq \frac{\eta_T^{(B)}}{T} + \eta_T^{(B)}\frac{w_+}{w_-}\sqrt{C_1\frac{\gamma_{TB}}{TB}}\right\}\right) \geq 1 - \delta \tag{6}$$

where for EST, $\eta_t^{(B)} = \nu_{t^*} + \zeta_t$ and for UBC, $\eta_t^{(B)} = 2(\beta_{t,1}^{(B)UCB})^{1/2}$, and $t^* = \underset{s\in[t]}{\operatorname{argmax}}\,\nu_s$.

*Proof.* See. Supp. Sec. A. $\square$

*Remark* 3.7. This theorem shows that, for the same kernel, the regret bound of LAW also enjoys the same asymptotic behavior as the regret bound of existing works[CBRV13, DKB14, KDK16].

*Remark* 3.8. Note that Thm. 3.6 is about a bound on $\frac{1}{T}R_T^{(B)}$ while the analysis in [DKB14, KKSP18] is to bound $\frac{1}{TB}R_{T,B}$ where $R_{T,B} = \sum_{t,b}r_{t,b}$. Since $\frac{1}{T}R_T^{(B)} \leq \frac{1}{TB}R_{T,B}$, bounding $\frac{1}{TB}R_{T,B}$ implies bounding $\frac{1}{T}R_T^{(B)}$. For the purpose of showing vanishing simple regret, both approaches are viable. Technically, two approaches require different treatments. See Supp. Subsec. A.2 for the discussion on the differences between two approaches.

*Remark* 3.9. The ratio $\frac{w_+}{w_-}$ in Thm. 3.6 determines how LAW balances between the quality and the diversity. If the ratio is large, then the acquisition value is more influential in Eq. 2. Otherwise, Eq. 2 is dominated by the determinant of the diversity gauge, and the diversity of the batch is more emphasized. The bound in Eq. 6 reveals the necessity of the upper bound of $\frac{w_+}{w_-}$. Without the upper bound, i.e. virtually considering the acquisition value only, the batch acquisition may result in non-vanishing regret. However, the bound is not tight enough considering the extreme case $\frac{w_+}{w_-} = 1$. Nonetheless, the necessity of the upper bound of $\frac{w_+}{w_-}$ guides how the weight function $w(\cdot)$ is set (See for details). Moreover, the benefit of considering the acquisition weight is supported by the experimental results (Sec. 5)

Note that $\eta_T^{(B)} = \mathcal{O}(\sqrt{\log(TB)})$ (See Supp.Subsec. A.3 for details). In Thm. 3.6, we need $\eta_T^{(B)} \cdot \sqrt{\frac{\gamma_{TB}}{TB}} \to 0$ to prove vanishing simple regret. We provide a bound for the maximum information gain $\gamma_T$ of a kernel on a finite space, which we use later to show the vanishing simple regret.

**Theorem 3.10.** *$K$ is a kernel on a finite set $\mathcal{X}$ ($|\mathcal{X}| < \infty$), $\sigma^2$ is the variance of the observation noise and $\Lambda = \{\lambda_n\}_{1,\cdots,|\mathcal{X}|}$ ($\lambda_n \geq \lambda_{n+1} \geq 0$) is the set of eigenvalues of the gram matrix $K(\mathcal{X},\mathcal{X})$. Then*

$$\gamma_T \leq \frac{1}{2}\min\{T \cdot \log\det(1 + \sigma^{-2}\max_{x\in\mathcal{X}}K(x,x)), |\mathcal{X}| \cdot \log(1 + \sigma^{-2}\lambda_{max}T)\} \tag{7}$$

*where $\lambda_{max}$ is the largest eigenvalue of $K(\mathcal{X},\mathcal{X})$.*

*Proof.* See. Supp.Subsec. B.1 $\square$

### 3.3 Position Kernel

Based on the comparative experiments in [ZSBB14] showing that the Position kernel outperforms others consistently,[2] we use the position kernel in our BBO on permutations

$$K(\pi_1,\pi_2|\tau) = \exp\left(-\tau\cdot\sum_i|\pi_1^{-1}(i) - \pi_2^{-1}(i)|\right).$$

The positive definiteness of the position kernel was empirically tested via simulation[ZSBB14] and has not been shown rigorously. Therefore, we show the positive definiteness of the position kernel and bound its eigenvalues.

**Theorem 3.11.** *The position kernel $K(\cdot,\cdot|\tau)$ defined on $S_N$ is positive definite and the eigenvalues of $K(X,X)$ where $X \subset \mathcal{X}$ lie between $\left(\frac{1-\rho}{1+\rho}\right)^N$ and $\left(\frac{1+\rho}{1-\rho}\right)^N$ where $\rho = \exp(-\tau)$.*

*Proof.* See Supp.Subsec. B.2 $\square$

---

[2]We also compared different kernels on regression tasks, including Kendall, Mallow [JV15], Hamming, Manhattan, Position [ZSBB14] and Neural Kernel Network (NKN) [SZW+18] using mentioned kernels as building blocks. The position kernel and NKN performs similarly the best. NKN uses the position kernel as a building block kernel which is attributed to the position kernel in the performance of NKN.

By utilizing the property of the position kernel, we provide a bound on the maximal information gain which is tighter than the one obtained in Thm. 3.10.

**Theorem 3.12.** *$K(\cdot, \cdot | \tau)$ is the position kernel defined on $S_N$, $\sigma^2$ is the variance of the observation noise, $\rho = \exp(-\tau)$ and, $D_{max} = (N^2 - (N \bmod 2))/2$.*

*Then*

$$\gamma_T \leq \frac{1}{2} \min\{A(T), |\mathcal{X}| \cdot \log(1 + \sigma^{-2}\lambda_{max}T)\}$$

*where $\lambda_{max}$ is the largest eigenvalue of $K(\mathcal{X}, \mathcal{X})$ and*

$$A(T) = \log(1 + \sigma^{-2}(1 + (T-1)\rho^{D_{max}})) + (T-1)\log(1 + \sigma^{-2}(1 - \rho^{D_{max}}))$$

*which is smaller than $T \cdot \log(1 + \sigma^{-2}\max_{x \in \mathcal{X}} K(x, x))$.*

*Proof.* See Supp.Subsec. B.3 □

*Remark* 3.13. When $\rho \in (0, 1)$ is close to one, i.e. $\log(1 + \sigma^{-2}(1 - \rho^{D_{max}})) \approx 0$, we can observe that even in the finite-time regime, the regret is almost sublinear since it is dominated by $\log(1 + \sigma^{-2}(1 + (T-1)\rho^{D_{max}}))$. In this case, the theorem provides a bound which is significantly tighter than the bound in Thm. 3.10 even in the finite-time regime.

*Remark* 3.14. If $\lambda_{max}$ is bounded, Thm. 3.10 can show the vanishing simple regret. For a kernel $K$ on a finite space $\mathcal{X}$, $\lambda_{max} \leq trace(K(\mathcal{X}, \mathcal{X})) < \infty$. Therefore, $\gamma_T = \mathcal{O}(log(T))$ for any kernel. However, considering the magnitude of $|\mathcal{X}|$ and $\lambda_{max}$ for large spaces, Eq. 7 is quite loose. $\lambda_{max}$ in Eq. 7 reflects kernel-dependent behavior of $\gamma_T$. Therefore, in Thm. 3.12 we bound $\lambda_{max}$ for a specific kernel and analyze further kernel-dependent non-asymptotic behavior.

The regret bounds of LAW are most informative in the asymptotic regime of large $T$. However, in Bayesian optimization where, typically, only a small number of evaluations can be afforded, the asymptotic bound may not be informative in terms of practical performance. In Sec. 5, we show that, in practice, LAW significantly outperforms other methods.

## 4 Related work

Most existing batch Bayesian optimization methods using Gaussian process surrogate models focus on continuous search spaces. Many of them are not applicable to combinatoral spaces because the algorithms use specific properties of Euclidean spaces, e.g, Euclidean distance [AFF10, GDHL16, WF16, WZJ16, KDK16, LYY+18], grid partitioning [WLJK17, WGKJ18], projection using Euclidean geometry [WCLF20]. The methods [SG15, KKSP18, GPL19] using Thompson sampling (TS) [Tho33, WBT+20], random feature [RR+07] or entropy search [HS12, HLHG14] require either closed-form expression of eigenfunctions or Choleksy decomposition of the gram matrix on all points in the search space. In general, a closed-form expression of eigenfunctions (RBF) is not available. For large combinatorial spaces, Choleksy decomposition is infeasible. LAW is a batch acquisition method applicable to general spaces including permutation spaces.

Recently, BO on combinatorial spaces has made significant progress for categorical variables [BP18, DBDF20, OTGW19, SRDM20, DSR+20]. However, relatively few works in Bayesian optimization have focused on permutations[ZSBB14, ZYLW19, BBG+20]. While existing works focus on the effect of the kernel on performance, our focus is to scale up Bayesian optimization on permutations via batch acquisition, which has not been studied in previous works.

The application of determinantal point processes (DPPs) to Bayesian optimization is not new. The use of DPP and the regret analysis on continuous search spaces[KDK16] is closely related to our work. We focus on optimization problems on permutations rather than continuous spaces and use acquisition weighted kernels in our DPP. We provide a regret bound, which includes the unweighted case as a special case. Moreover, we show the behavior of the information gain of the position kernel, which, in turn, helps to understand the behavior of BO on permutations.

The idea of using weighted kernels was investigated in DPP [KT10, KT12], also recently in the context of active learning[BWAS19] and more recently in architecture search [NLYAO21]. In addition to the use of the acquisition weights, we provide a regret analysis and the bound on the information gain of the position kernel for BO on permutations.

In existing works on regret analysis of batch Bayesian optimization, the cumulative regret is analyzed as an end goal[DKB14] and as a medium to show vanishing simple regret[KKSP18]. On the other

hand, we analyze the batch cumulative regret not the cumulative regret (see Def. 3.3 and remarks thereafter). The batch cumulative regret is analyzed in [CBRV13] but without the acquisition weight. More detailed discussion on the difference among all these analyses is provided in Supp.Subsec. A.2.

# 5 Experiments

We empirically demonstrate the benefit of LAW on many optimization problems on permutations.[3]

In all Gaussian process (GP) based BO including baselines, we use the position kernel (see. Subsec. 3.3). At each round, evaluation outputs are normalized. GP surrogate models are trained with output normalized evaluation data by optimizing the marginal likelihood until convergence with 10 different random initializations. We use the Adam optimizer[KB14] with default PyTorch[PGC+17] settings except for the learning rate of $0.1$.

When the optimization is performed on a single permutation variable, for example in greedy optimization, hill climbing is used until convergence and the neighbors are defined as the set of permutations obtainable by swapping two locations.

## 5.1 Weight function

The motivation of the acquisition weight is to promote the quality of the queries in the batch by using acquisition weights. In order to reflect this motivation, the weight function should be monotonically increasing. The better the quality (acquisition value) is, the larger the batch acquisition objective is.

In Eq. 2, the batch acquisition objective is factorized into the DPP with the diversity gauge and the product of weights, thus the weight function should be positive to prevent the product of an even number of large negative values becomes a large positive value. Also, in Eq. 2, zero weight value nullifies the diversity component, thus the weights function is required to be nonzero.

In Thm. 3.6, for vanishing regret, the ratio $\frac{w_+}{w_-}$ should be upper-bounded. Not only it facilitates the proof, but the upper bound is also intuitively appealing because we do not want to overly emphasize the quality of the queries. We want to balance quality and diversity. Weight functions with an unbounded ratio may erase the diversity consideration.

Combining the rationale behind LAW and its regret analysis, we set the weight function to be, monotonically increasing, positive valued, bounded below, bounded above.

## 5.2 Combinatorial Optimization Benchmarks

We consider LAW with two acquisition functions[4], EST[WZJ16] and EI[JSW98], LAW-EST and LAW-EI. Even though the regret bound of LAW-EI is not provided in Thm. 3.6, we include LAW-EI because EI is the most popular acquisition function and this reveals the effect of the acquisition weights with different acquisition functions. We use the sigmoid $w(a) = 0.01 + 0.99(1 + \exp(-0.2 \cdot a))^{-1}$ for LAW-EST and $w(a) = 0.01 + a$ for LAW-EI.[5]

The baselines are q-EI, q-EST[GLRC08], BUCB[DKB14], DPP-MAX-EST, DPP-SAMPLE-EST[KDK16][6] and MACE-UCB, MACE-EST[LYY+18][7]. Even though the original names of the baselines are used to emphasize their batch acquisition strategy, all baselines use the position kernel. Hence, the batch acquisition strategy is the only differentiating factor among baselines and LAW(ours). Note that DPP-MAX-EST [KDK16] corresponds to LAW-EST with $w(\cdot) \equiv$ const., i.e., no acquisition weight.

---

[3]The code is available at `https://github.com/ChangYong-Oh/LAW2ORDER`

[4]The $\beta_t$ in UCB balancing between exploitation and exploration increases as the size of the search space increases in the finite search space case[SKKS09]. In the experiments, due to the size of the search space, GP-UCB virtually becomes the predictive variance. Thus we exclude LAW-UCB.

[5]LAW-EI is included to check the influence of different acquisition functions despite the lack of regret analysis. Therefore, the weight function is chosen to prevent zero values from numerical truncation.

[6]PE[CBRV13] is equivalent to DPP-MAX-UCB[KDK16]. Since on continuous problems DPP-MAX-EST outperforms DPP-MAX-UCB[KDK16], we exclude PE.

[7]The MACE requires multi-objective optimization on permutations. We use NSGA-II[DPAM02] in Pymoo[BD20]. MACE-UCB uses the original set of acquisition functions: PI, EI and UCB[LYY+18], while MACE-EST replaces UCB with EST.

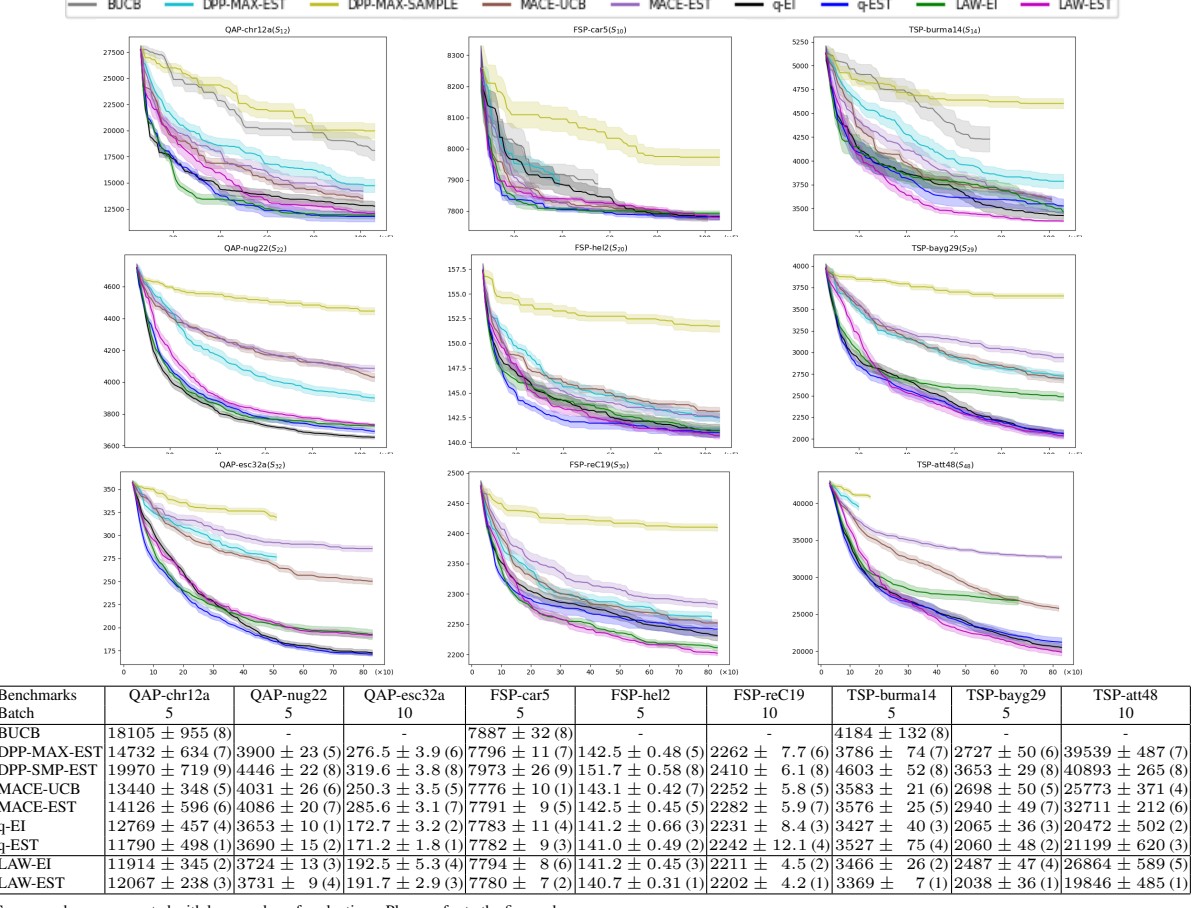

| Benchmarks | QAP-chr12a | QAP-nug22 | QAP-esc32a | FSP-car5 | FSP-hel2 | FSP-reC19 | TSP-burma14 | TSP-bayg29 | TSP-att48 |
|---|---|---|---|---|---|---|---|---|---|
| Batch | 5 | 5 | 10 | 5 | 5 | 10 | 5 | 5 | 10 |
| BUCB | $18105 \pm 955$ (8) | - | - | $7887 \pm 32$ (8) | - | - | $4184 \pm 132$ (8) | - | - |
| DPP-MAX-EST | $14732 \pm 634$ (7) | $3900 \pm 23$ (5) | $276.5 \pm 3.9$ (6) | $7796 \pm 11$ (7) | $142.5 \pm 0.48$ (5) | $2262 \pm 7.7$ (6) | $3786 \pm 74$ (7) | $2727 \pm 50$ (6) | $39539 \pm 487$ (7) |
| DPP-SMP-EST | $19970 \pm 719$ (9) | $4446 \pm 22$ (8) | $319.6 \pm 3.8$ (8) | $7973 \pm 26$ (9) | $151.7 \pm 0.58$ (8) | $2410 \pm 6.1$ (8) | $4603 \pm 52$ (8) | $3653 \pm 29$ (8) | $40893 \pm 265$ (8) |
| MACE-UCB | $13440 \pm 348$ (5) | $4031 \pm 26$ (6) | $250.3 \pm 3.5$ (5) | $7776 \pm 10$ (1) | $143.1 \pm 0.42$ (7) | $2252 \pm 5.8$ (5) | $3583 \pm 21$ (6) | $2698 \pm 50$ (5) | $25773 \pm 371$ (4) |
| MACE-EST | $14126 \pm 596$ (6) | $4086 \pm 20$ (7) | $285.6 \pm 3.1$ (7) | $7791 \pm 9$ (5) | $142.5 \pm 0.45$ (5) | $2282 \pm 5.9$ (7) | $3576 \pm 25$ (5) | $2940 \pm 49$ (7) | $32711 \pm 212$ (6) |
| q-EI | $12769 \pm 457$ (4) | $3653 \pm 10$ (1) | $172.7 \pm 3.2$ (2) | $7783 \pm 11$ (4) | $141.2 \pm 0.66$ (3) | $2231 \pm 8.4$ (3) | $3427 \pm 40$ (3) | $2065 \pm 36$ (3) | $20472 \pm 502$ (2) |
| q-EST | $11790 \pm 498$ (1) | $3690 \pm 15$ (2) | $171.2 \pm 1.8$ (1) | $7782 \pm 9$ (3) | $141.0 \pm 0.49$ (2) | $2242 \pm 12.1$ (4) | $3527 \pm 75$ (4) | $2060 \pm 48$ (2) | $21199 \pm 620$ (3) |
| LAW-EI | $11914 \pm 345$ (2) | $3724 \pm 13$ (3) | $192.5 \pm 5.3$ (4) | $7794 \pm 8$ (6) | $141.2 \pm 0.45$ (3) | $2211 \pm 4.5$ (2) | $3466 \pm 26$ (2) | $2487 \pm 47$ (4) | $26864 \pm 589$ (5) |
| LAW-EST | $12067 \pm 238$ (3) | $3731 \pm 9$ (4) | $191.7 \pm 2.9$ (3) | $7780 \pm 7$ (2) | $140.7 \pm 0.31$ (1) | $2202 \pm 4.2$ (1) | $3369 \pm 7$ (1) | $2038 \pm 36$ (1) | $19846 \pm 485$ (1) |

Some numbers are reported with less number of evaluations. Please refer to the figure above.

Table 1: Permutations Benchmarks (Mean ± Std.Err.(Rank) over 15 runs)

Note that, due to the reasons discussed in Sec. 4, existing works based on Thompson sampling or the properties of Euclidean space are excluded from the baselines.

We consider three types of combinatorial optimization on permutations, Quadratic Assignment Problems(QAP), Flowshop Scheduling Problems(FSP) and Traveling Salesman Problems(TSP) (See Supp.Subsec. F.3 for data source).

For each benchmark, all methods share 5 randomly generated initial evaluation data sets of 20 points and for each initial evaluation data set, each method is run three times — 15 runs in total.

DPP-MAX-EST uses the position kernel as LAW-EST, this is equivalent to LAW-EST without the acquisition weight, i.e. $w(a) = 1$. By comparing LAW-EST with DPP-MAX-EST, we can directly evaluate the benefits of using the acquisition weight.

As shown in Tab. 5.2, LAW-EI, LAW-EST, q-EI and q-EST are in top four except for FSP-car5 and TSP-att48. LAW-EST performs the best on FSP and TSP while q-EI or q-EST perform the best on QAP. Along with the experiment on the structure learning (Subsec. 5.3), we conjecture that QAP has a certain structure more friendly to q-EI and q-EST. Also LAW-EI exhibits comparable performance with q-EI and q-EST while outperforming other baselines, which supports the benefit of the acquisition weight. In terms of the average rank over all benchmarks, LAW-EST performs the best with the average rank of $1.89$ against q-EST ($2.44$) and q-EI ($2.78$). Overall, among the baselines, LAW-EST exhibits stable and competitive performance across different benchmarks.

### 5.2.1 Comparison to the local penalization (LP)

Two additional variants of LAW, LAW-PRIOR-EST and LAW-PRIOR-EI, are also compared (Supp. Sec. G), which use the prior covariance function as the diversity gauge, $L = K$, of $L_t^{AW}$. These variants do not use evaluation data in the diversity gauge.

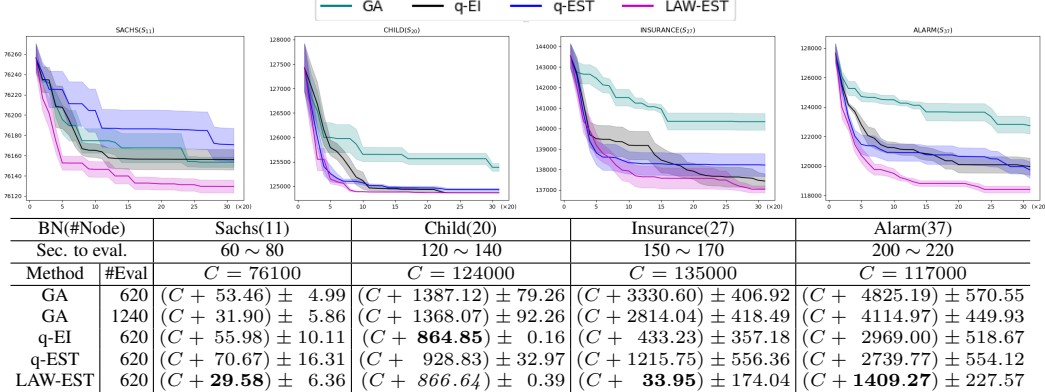

| BN(#Node) | | Sachs(11) | Child(20) | Insurance(27) | Alarm(37) |
|---|---|---|---|---|---|
| Sec. to eval. | | $60 \sim 80$ | $120 \sim 140$ | $150 \sim 170$ | $200 \sim 220$ |
| Method | #Eval | $C = 76100$ | $C = 124000$ | $C = 135000$ | $C = 117000$ |
| GA | 620 | $(C + 53.46) \pm 4.99$ | $(C + 1387.12) \pm 79.26$ | $(C + 3330.60) \pm 406.92$ | $(C + 4825.19) \pm 570.55$ |
| GA | 1240 | $(C + 31.90) \pm 5.86$ | $(C + 1368.07) \pm 92.26$ | $(C + 2814.04) \pm 418.49$ | $(C + 4114.97) \pm 449.93$ |
| q-EI | 620 | $(C + 55.98) \pm 10.11$ | $(C + \mathbf{864.85}) \pm 0.16$ | $(C + 433.23) \pm 357.18$ | $(C + 2969.00) \pm 518.67$ |
| q-EST | 620 | $(C + 70.67) \pm 16.31$ | $(C + 928.83) \pm 32.97$ | $(C + 1215.75) \pm 556.36$ | $(C + 2739.77) \pm 554.12$ |
| LAW-EST | 620 | $(C + \mathbf{29.58}) \pm 6.36$ | $(C + \mathit{866.64}) \pm 0.39$ | $(C + \mathbf{33.95}) \pm 174.04$ | $(C + \mathbf{1409.27}) \pm 227.57$ |

Table 2: Negative log NML minimization for the structure learning (Mean $\pm$ Std.Err. over 5 runs)

Interestingly, LAW-PRIOR-EST and LAW-PRIOR-EI resemble LP[GDHL16] (Supp. Sec. C), and thus this allows an indirect comparison to LP which is not applicable to combinatorial spaces.[8]

These variants using the prior covariance function performs worse than LAW using the posterior covariance function, which is natural since using more data for the diversity gauge enhances the performance. More importantly, LAW-PRIOR-EST and LAW-PRIOR-EI outperform DPP-MAX-EST which uses the posterior covariance function without the acquisition weight, which supports that the acquisition weight is key in the performance improvement.

Based on the empirical analysis above, we choose LAW-EST as our final recommendation, which we call LAW2ORDER.

### 5.3 Structure Learning

We apply LAW2ORDER to the score-based structure learning problem[DM17]. Existing score-based methods assume a computationally amenable structure of the score to be optimized (decomposability)[KF09, SGG19]. Distinctively our approach does not necessitate the decomposability of the score to be optimized.

We consider the NML score as below

$$S_{NML}(\mathcal{G}, \mathcal{D}) = -\log p_{BN}(\mathcal{D} \,|\, \mathcal{G}, \hat{\theta}_{ML}(\mathcal{G}, \mathcal{D})) + REG_{NML}(\mathcal{G}, |\,\mathcal{D}\,|) \qquad (8)$$

where $p_{BN}(\cdot \,|\, \mathcal{G}, \theta)$ is the density of Bayesian Network (BN) with DAG $\mathcal{G}$ and the parameter $\theta$, and $REG_{NML}$ is the normalized marginal likelihood (NML) which is a complexity measure from the minimum description length principle[GG07]. NML is not decomposable, and thus the methods assuming a decomposable score are not applicable. Since it is infeasible to compute NML exactly and we resort on MC estimate, the NML score evaluation is noisy. In addition to not being decomposable, noisy evaluation also makes existing methods inapplicable to the NML score objective. For more details of NML and its MC estimate, see Supp.Sec. E.

Similarly to [SWU17, RU18] we search over permutations specifying topological order of DAGs and the existence of edges is determined by the conditional independence test. In addition to q-EI and q-EST performing well on the benchmarks, we compare LAW2ORDER with the genetic algorithm (GA), which is one of the most popular choices for optimization problems on permutations including TSP[Pot96].

We generated 5 sets of 20 random initial points. LAW2ORDER, q-EI and q-EST is run on each of these 5 sets using a batch size 20. Assuming the same resource constraint (at most 20 evaluations in parallel), GA generates 20 off-springs in each generation. GA is also run 5 times with a population size of 100 points using Pymoo[BD20]. The first 20 points of each initial population in the 5 runs are equal to the 20 initial points used in LAW2ORDER, q-EI and q-EST. Even though the real deployment of Bayesian optimization assumes that the cost of evaluation is expensive enough to render the time to acquire new batches negligible, as a stress test, we allowed twice the evaluation budget for GA.

---

[8]LAW variants use the kernel of the GP surrogate model as the diversity gauge which is more guided by data while LP uses the local penalizer which is heuristically designed. We expect that this distinction will still make a difference on the performance.

On data generated from 4 real-world BNs[Scu10, SGG19], the results are reported in Tab. 2. LAW2ORDER outperforms q-EI and q-EST with a significant margin except for Child where all three find the putative optimum quickly. Still, in Child, LAW2ORDER finds a point of negligible differences with the putative optimum the most quickly. Also, except for Sachs, we observe that the performance gap increases as the permutation size (the size of the search space) increases. On this realistic problem, our argument that LAW2ORDER is stable and efficient batch acquisition method on permutations is reinforced.

In comparison with GA, we consider GA(620) with the same evaluation budget and GA(1240) with twice large evaluation budget. LAW2ORDER dominates GA(620) in all problems with a significant margin. Even compared with GA(1240), LAW2ORDER significantly outperforms except for Sachs which has one of the smallest search spaces (See Tab. 2 and Supp. Sec. G). Contrary to our expectation that BO has the sample efficiency higher than GA, GA(620) outperforms q-EI and q-EST on Sachs. LAW2ORDER shows robust performance even in the problem where the performance of q-EI and q-EST is degraded.

Together with the experiments on the benchmarks, LAW2ORDER is shown to be a robust batch acquisition method on permutations. Moreover, promoting the diversity in batches while taking into account the acquisition weight appears more beneficial with larger batch size (20) as shown in the structure learning experiment.

# 6 Conclusion

In this paper we have focused on combinatorial optimization problems over permutations where each evaluation is assumed to be expensive. This class of problems has many interesting applications, ranging from chip design (where we wish to place cells while minimizing area and wire-length), warehouse optimization (where we need to order the retrieval of items from a warehouse using a robot), neural architecture search and so on. In spite of its practical significance, BO on permutations is under-explored in contrast to the recent progress on combinatorial BO with categorical variables.

In response to this, we have proposed a batch Bayesian optimization algorithm on permutations, LAW2ORDER, which uses an extension of the determinantal point processes with the acquisition weighted kernel. This allows the search process over the surrogate function to make optimal use of all parallel available computational resources and be guided by both the expected objective value and its posterior uncertainty.

On the theory side we offer a regret analysis, which shows that the regret bound of LAW enjoys the same asymptotic behavior as existing methods. On the empirical side, we show that LAW variants consistently exhibit competitive performance on a wide range of combinatorial optimization tasks, including a challenging structure learning problem.

From these we conclude that the acquisition weights are indeed a key factor in the success of the proposed method, and that the performance gains increase for large batch sizes.

LAW is applicable to general search spaces for which a kernel can be defined. We leave the exploration of our method to applications outside searching over permutations for future work.

## 6.1 Limitations

LAW achieves improved sample efficiency in the sense that the quality of batches from LAW is maintained for large batch sizes. However, due to the sequential nature of the greedy maximization of LAW objective, its computational complexity is linear with respect to the batch size. Especially for large permutation spaces, this may be a nonnegligible cost. We hope that the sample efficiency of LAW is complemented by the computational efficiency allowing massive parallelization.

Even though the regret bound in Thm. 3.6 describes the effect of the acquisition weight, as mentioned in Rmk. 3.9, it only sheds light on the demerit of excessive emphasis on the acquisition weights but it does not detect the demerit of considering the diversity only. We hope our work inspires the regret bound for acquisition methods taking into account properties other than diversity.

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
