# OpenReview forum: "Batch Bayesian Optimization on Permutations using the Acquisition Weighted Kernel"
_NeurIPS.cc/2022/Conference — NeurIPS 2022 Accept_

### Official Review · Reviewer_CuLB · 2022-07-09

**Rating:** 5
**Confidence:** 2
**Soundness:** 3 good
**Presentation:** 3 good
**Contribution:** 3 good

**Summary:**

This paper provides a novel batch selection method, LAW, for bayesian optimisation over permutations. The authors give theoretical regret bounds and compare LAW against some baselines on three combinatorial optimisation problems.


**Questions:**

How does the computational cost of LAW compare to the baselines?


**Limitations:**

I don't see any potential negative societal impact of this work. The authors did not address limitations of their work.


**Strengths And Weaknesses:**

The introductory and background sections provide good motivation and compactly cover prerequisites for the rest of the paper. The theoretical results from section 3 are put into context, with their significance and their relation to other work clearly explained by remarks within section 3. The paper as a whole is generally well written.

I'm not convinced by the claim at the end of section 3 that LAW "significantly outperforms other methods". The results in section 5 appear to show that LAW is competitive with the baselines, but do not show a definitive advantage over the best competing baselines. I do, however, think that the performance demonstrated in section 5 is sufficiently good to justify publication.

I don't feel confident in assessing how novel LAW is, or how significant it is in the context of combinatorial bayesian optimisation.

---

> ### Author Response · Authors · 2022-08-02
> **Response to the preliminary review from Reviewer CuLB**
>
> We thank for the reviewer's constructive feedback. We provide our answers to the questions raised.
>
> ### Experiment result interpretation.
>
> We want to stress that the benefit of the method is more noticeable in the structure learning experiment where we use a much larger batch size and we compare it with the baselines which may seem to perform similarly well on the benchmarks. This bigger benefit with the larger batch sizes is noteworthy considering the purpose of the batch acquisition trading computes resources with wall-clock run time.
>
> We admit that on three benchmarks, some baselines are as competitive as our method. Therefore, we will tone down our claim appropriately.
>
>
> ### Computational Cost / Limitations
>
> All methods using greedy optimization for the batch acquisition function optimization except for MACE family have similar run-time in the batch acquisition. MACE family using multi-objective genetic algorithm is far faster in run-time.
> Even though this computational cost in the batch acquisition optimization is common in all DPP-based methods, we admit that this may be one of the limitations, especially when the method is applied to large permutation spaces, such as $S_{50}, S_{100}$.
> We appreciate the reviewer’s acute critique, we will cover this properly in the limitation in the revised version.
>
>
> ### Novelty in the context of combinatorial Bayesian optimization
>
> The permutation space has many distinctive properties compared with discrete & combinatorial spaces such as super-exponential growth and dependency among variables. In spite of such differences, Bayesian optimization on permutations is relatively new. Many existing batch acquisitions are not applicable to permutation spaces because they rely on the specific properties of Euclidean spaces. In response to the more desperate need for batch acquisition on such super-exponential spaces, we propose a batch acquisition balancing between quality and diversity. Moreover, to our knowledge, our approach is the first regret analysis taking into account not only diversity but also the quality of queries.
>
> For references and other details, we kindly refresh the reviewer's attention to Related Work.

---

> ### Comment · Area_Chair_jt37 · 2022-08-08
> **Interacting with the rebuttal**
>
> Dear reviewer,
>
> Can you read the author's rebuttal, check if it addresses your concerns, and react to it?
>
> It is important to acknowledge this work by the authors and to respect it.
>
> Best, AC

---

### Official Review · Reviewer_DqJ6 · 2022-07-10

**Rating:** 6
**Confidence:** 3
**Soundness:** 3 good
**Presentation:** 3 good
**Contribution:** 3 good

**Summary:**

This paper studies the problem of batch Bayesian optimization for combinatorial problems on permutations. The main idea is determinantal point processess using the acquisition weighted kernel. The proposed method LAW2ORDER is applied to solve permutation problems using Bayesian optimization. The paper provides theoretical guarantee that the batch cumulative regret is sublinear. Extensive evaluations on several standard combinatorial problems and a structure learning task demonstrate the effectiveness of LAW2ORDER.


**Questions:**

- It in unclear how to get or set weight function during joint optimization. Could you elaborate more on this? Is setting weight function problem-dependent?

I would like to hear more justifications on these from the authors.

**Limitations:**

- How to set the weight function is unclear. Therefore, it is unclear how to use it in practice without a principled way to set it.
- No code released.


**Strengths And Weaknesses:**

Strengths:

- Study an important problem of applying batch Bayesian optimization to solve permutation problems.
- Good theoretical analyses.
- Comprehensive experimental evaluation.
- Writing is easy to follow.

Weaknesses:

- No code released.
- Unclear how to set the weight function.

Originality:

The originality of the paper is two folds. First, it introduces a new efficient batch acquisition method based on DPP using the acquisition weighted kernel. Second, it applies LAW to solve permutation problems.


Quality:

The problem is well defined and the proposed method is complete. The algorithm is not complicated which makes the approach accessible to potentially wide audience.

Clarity:

The writing is clear and easy to follow.

Significance:

The paper studies a very important but rarely studied problem in solving permutation problems in batch Bayesian optimization.

---

> ### Author Response · Authors · 2022-08-02
> **Response to the preliminary review from Reviewer DqJ6**
>
> We thank for the reviewer's constructive feedback. We provide our answers to the questions raised.
>
> ### How to set the weight function
>
> The basic motivation is to promote the quality of the queries in the batch by using acquisition weights. In order to reflect this motivation, the weight function should be monotone, the better the quality (acquisition value) is, the larger the batch acquisition objective is.
>
> In eq (2), the acquisition weighted kernel DPP is factorized as a product of ordinary DPP and the product of weights, thus the weight function should be positive to prevent the product of an even number of large negative values becomes a large positive value.
> Also, in eq (2), zero weight value can zero out the diversity component, the weights function is also required to be nonzero.
>
> In the proof of the regret bound, the weight function is also required to be upper-bounded. Not only for accommodating the proof, but this is also intuitively appealing because we don’t want to overly emphasize the quality of the queries but we want to balance between the quality and the diversity. Unbounded weight functions may erase the diversity consideration.
> The boundedness of the weight function is not only justified by the facilitation of the proof but also aligned with the intuition behind the acquisition weight.
>
> With these conditions, positive valued, bounded below, bounded above, we chose the acquisition function using squashing functions e.g. for the method with a theoretical guarantee on the bound (LAW-EST).
>
> The addition of a small positive number is just for preventing numerical instability and preserving the positivity of the batch acquisition function. With some preliminary runs on QAP-chr12a, we observed the acquisition values and determined the weights function so that acquisition weights lie between $[\epsilon, 1]$ $\epsilon > 0$. In all experiments, the same weight function is used.
>
> With $w_+ = \sup w(a(\cdot))$ and $w_- = \inf w(a(\cdot))$, the ratio $\frac{w_+}{w_-}$ determines how we balance between the quality and the diversity. If the ratio is large then the quality is a more influential differentiating factor, otherwise, the quality affects less in LAW batch acquisition function. As long as the weight function satisfies the conditions, monotone, positive-valued, bounded-below, and bounded-above, we conjecture that this ratio is more influential than the specific shape of the weight function.
>
> At least in all of our experiments including follow-up real-world applications, the same weight function is consistently effective. We conjecture that the ratio $\frac{w_+}{w_-}$ can be a knob to control how to balance between the quality and the diversity and that problem-specific performance improvement can be obtained. Nonetheless, based on its consistent performance, we recommend our proposed weight function or simple variants of it as a starting point.

---

### Official Review · Reviewer_9inD · 2022-07-19

**Rating:** 6
**Confidence:** 4
**Soundness:** 2 fair
**Presentation:** 2 fair
**Contribution:** 2 fair

**Summary:**

The paper proposes a new approach for BO to solve expensive permutation problems. The idea is to use batch BO to acquire a batch of data points and evaluate them in parallel. To this end, the paper proposes the use of determinantal point process (DPPs) combined with a weighting scheme to construct the batch acquisition function. Besides, the position kernel is suggested to use within the acquisition function to enhance the performance of the proposed methods. Theoretical analysis is conducted to evaluate the efficacy of the proposed approach. Numerical experiments on three combinatorial optimization benchmarks on permutations, and the structure learning problem.

**Questions:**

Apart from my comments in the Weaknesses section that the authors can response, I have the following questions:

1.	Is the Acquisition Weighted kernel a valid kernel? Could you also give rigorous explanation for your answer?
2.	Are the lambda_{max} values in Theorems 3.9 and 3.12 upper bounded by constants when T goes to infinity?


**Limitations:**

I think the paper does not mention much about the limitations or the societal impacts of their proposed approach.

**Strengths And Weaknesses:**

Strengths: I think developing new BO methods for permutation problems is important and is worth to investigate. I also think the idea of using DPP is interesting. The authors also put effort on analysing the theoretical properties of their proposed approach.

Weaknesses: Although I like the general idea of using DPP, I found there are various issues with the current version of the paper. Please see my detailed comments as follows.

•	The paper specifically targets the permutation problems, but I don't see how this permutation property is incorporated into the design of the proposed acquisition function (except the fact that batch BO is used so that we can evaluate multiple data points parallel and thus can avoid the issue of large search space for permutation problems).

•	Even though the paper provides various theoretical analysis, but these analyses seem not to be rigorous, and might not really help to answer the analytical properties of the proposed approach.

       o	The Acquisition Weighted Kernel L^{AW} defined in Line 122 seems to not be a real (valid) kernel? As it depends on any acquisition function a(x), so it seems impossible to me that it is a valid kernel for all cases. Besides, this seems to be like a component of the proposed acquisition function, rather than to be called a "kernel".
      o	The regret analysis in Theorem 3.6 depends on the maximum information gain \gamma_T, but this \gamma_T value is not properly bounded in Theorem 3.9. Theorem 3.9 only shows that \gamma_T is smaller than a function of \lambda_{max} but there is no guarantee that \lambda_{max} is bounded when T goes to infinity. This is a key analysis in any BO analysis. In the literature, only several kernels have been shown that their \lambda_{max} is bounded when T goes to infinity.
      o	Again, same problem with Theorem 3.12, is there any guarantee that the \lambda_{max} of the position kernel is upper bounded?

•	The performance of the proposed approach on the permutation optimization problems are not that good though (Section 5.1). LAW-EI performs pretty bad, much worse than other baselines in various cases, while LAW-EST is on par with other baselines and only performs well in one problem. Besides, I don't understand why LAW-UCB is not added as one of the baselines. The justification regarding the size of search space does not seem reasonable to me.

•	Besides, the experiments seem not too strong and fair to me. I don't understand why all the baselines use the position kernels, why don't we use the default settings of these baselines in the literature? Besides, it seems like some baselines related to BO with discrete & categorial variables are missing. The paper also needs to compare its proposed approach with these baselines.

---

> ### Author Response · Authors · 2022-08-02
> **Response to the preliminary review from Reviewer 9inD - Kernel & Regret Bound**
>
> We thank for the reviewer's constructive feedback. Here we provide our answers to the questions on the theoretical part.
>
> ### Positive-definiteness of the acquisition weighted kernel.
>
> **Definition** [Wiki: Positive-definite kernel]: (https://en.wikipedia.org/wiki/Positive-definite_kernel) Positive-definite kernel
>
> Let $\mathcal{X}$ be a nonempty set, sometimes referred to as the index set. A symmetric function
> $K:\mathcal{X} \times \mathcal{X} \to \mathbb{R}$ is called a positive-definite (p.d.) kernel on $\mathcal{X}$ if
> $$\sum_{i=1}^{n} \sum_{j=1}^{n} c_{i} c_{j} K(x_{i}, x_{j}) \geq 0$$
> holds for any $x_{1}, \dots, x_{n} \in \mathcal{X}$, given $n \in \mathbb{N}, c_{1}, \dots, c_{n} \in \mathbb{R}$.
>
> Consider a weighted kernel $K_{w}(x_{i}, x_{j}) = w(x_{i}) K(x_{i}, x_{j}) w(x_{j})$. Here we assume an arbitrary weight function $w(\cdot)$.
> Plugging in, we have
> $$\sum_{i=1}^{n} \sum_{j=1}^{n} c_{i} K_{w}(x_{i}, x_{j}) c_{j} = \sum_{i=1}^{n} \sum_{j=1}^{n} c_{i} w(x_{i}) K(x_{i}, x_{j}) w(x_{j}) c_{j}$$
> Then the positive-definiteness of $K_{w}(\cdot,\cdot)$ follows from the positive-definiteness of $K(\cdot, \cdot)$ with $c_{i}' = c_{i} w(x_{i})$.
> Since $c_{1}, \cdots, c_{n}$ in the definition are arbitrary, the definition still holds with $c_{i}'$ depending on $x_{i}$. Simply speaking, the definition with $c_{i}$ and $K_{w}(\cdot,\cdot)$ is equivalent to the definition with $c_{i}/w(x_{i})$ and $K_{w}(\cdot,\cdot)$.
>
>
>
> ### Condition on $\lambda_{max}$ for sublinear regret
>
> In our bound, the boundedness of $\lambda_{max}$ is a sufficient condition, not the necessary condition for sublinear regret since $\lambda_{max}$ appears inside the log. As long as $\lambda_{max}$ has a growth rate slower than exponential, e.g. polynomial, it has the sublinear regret. For example, with $\lambda_{max} \sim T^{k-1}$, $\frac{1}{T} \log(1 + \sigma^{-2} T^k) \to 0$ as $T \to \infty$ for $k \in \mathbb{N}$. In general, the sublinearity of the regret still holds with $\lambda_{max}$ of polynomial growth.
>
> On finite spaces, gram matrices of nonrepeated points $[K(x,y)]_{x,y \in D \subset S_N}$
>
> are principal submatrices of the gram matrix of all points $[K(x,y)]_{x,y \in S_N}$.  Based on this, the Theorem 3.11 shows that on the space of permutations of length $N$, $S_N$, all eigenvalues lie in $\Bigg[\Big(\frac{1-\rho}{1+\rho}\Big)^N, \Big(\frac{1+\rho}{1-\rho}\Big)^N\Bigg]$ where $\rho = \exp(-\tau)$, which is independent with $T$.
>
> When a gram matrix is constructed on a set of points with at most $R$ repetitions, this can be considered as a principal submatrix of the gram matrix of the set in which every point repeats $R$ times $[K(x,y)]_{x,y \in S_N} \otimes (1_R 1_R^T)$.
>
> Therefore with at most $R$ repetitions, the eigenvalues of gram matrix lie in $\Bigg[0, R\Big(\frac{1+\rho}{1-\rho}\Big)^N\Bigg]$. Since $R \le T$, even when the gram matrix is constructed on a set with repeated points, the eigenvalues of the gram matrix are bounded above by $R\Big(\frac{1+\rho}{1-\rho}\Big)^N$. Therefore, $\frac{1}{T} \log(1 + \lambda_{max}T) \le \frac{1}{T} \log \Big(1 + R \Big(\frac{1+\rho}{1-\rho}\Big)^N T \Big) \to 0$.
>
> Even though the boundedness is not essential for the sublinear regret, we appreciate the feedback which refreshes attention to the necessity of the growth of $\lambda_{max}$.

---

> > ### Comment · Reviewer_9inD · 2022-08-06
> > **Thank you for the authors' response**
> >
> > Dear authors,
> >
> > Thank you for your effort in responding to my concerns. The response have addressed some of my concerns, however, some concerns still remain. Please find in the below my response:
> >
> > 1.	Thank you for the explanation of the positive-definiteness of the acquisition weighted kernel. The authors' explanation make sense to me, although note that, in the explanation, the term \omega(x_i) should be \omega(a(x_i)).
> >
> > 2.	In Theorem 3.9, it's true that \lambda_max does not need to be upper bounded by constant when T goes to infinity (my bad for saying this), but it is true that \lambda_max has to have a growth rate slower than exponential (as the authors discussed). In Remark 3.10, it states that Theorem 3.9 shows vanishing simple regret – which is not correct. If I understand correctly, Theorems 3.8 and 3.9 are proven for all types of kernels K (right?) and the Remark 3.10 is applied for all kernels (right?), but we do not have that \lambda_max has a growth rate slower than exponential for all kernels. In summary, my point is that, using Theorems 3.6 and 3.9 does not show vanishing simple regret as in Remark 3.10; this vanishing simple regret property depends on the chosen kernel itself.
> >
> > 3.	Now after I dig deeper into the proof, I think Theorems 3.11 might be correct but all the notations of N, S_N, |X|, \pi make it extremely difficult to verify the correctness of the theorem. Similar for Theorem 3.12, the authors should consider making the notations and the writing of the proof to be better in the revised version. Note that now I don't have much concern with Theorem 3.12 anymore, but again, this is more based on my general understanding/gut feelings, not that I can clearly verify the proof of Theorems 3.11 and 3.12 step by step.
> >
> > 4.	Comparison with other kernels: I understand that the focus is on the batch acquisition. In the rebuttal, the authors mentioned that based on the previous results and the authors' preliminary runs comparing different kernels, the position kernel is picked as it is the best performer. But this is not demonstrated in the paper (please correct me if this is actually shown in the paper in case I missed it). As a reader, I have to see the results proving that in this case, the position kernel is a good choice for evaluation. Otherwise, it's hard to confirm the effectiveness of the proposed and the baseline acquisition functions.

---

> > > ### Author Response · Authors · 2022-08-07
> > > **Response to the Reviewer 9inD on remaining concerns**
> > >
> > > Thanks for responding to our rebuttal with constructive feedback.
> > >
> > > ### On the Remark 3.10
> > > As the reviewer pointed out, the sublinear regret depends on the properties of $\lambda_{max}$. We agree that the sublinearity of the simple regret is shown by combining Thm 3.6, Thm 3.9, Thm 3.11, and Thm 3.12. Thus, any remarks on the sublinearity should come after Thm 3.11 and Thm 3.12. We will fix the Remark 3.10 accordingly and relocate the remark on the sublinearity after stating properties of $\lambda_{max}$ (Thm 3.11 and Thm 3.12). We appreciate the reviewer’s detailed feedback on this. Also, we will clarify further the proofs in the revised version reflecting the reviewer’s feedback.
> > >
> > > ### Comparison with other kernels
> > > One of reasons we chose the position kernel is supported empirically in the paper [Distance Measures for Permutations in Combinatorial Efficient Global Optimization](https://www.researchgate.net/profile/Thomas-Bartz-Beielstein/publication/266736370_Distance_Measures_for_Permutations_in_Combinatorial_Effcient_Global_Optimization/links/61253f65169a1a010324ac57/Distance-Measures-for-Permutations-in-Combinatorial-Effcient-Global-Optimization.pdf).
> > > In this paper, they compared different types of kernels and the position kernel was shown to perform well consistently. Since our preliminary runs are just reconfirmation of their claim, we will make it clear that the strength of the position kernel is supported by the above paper. Even though we explained why we chose the position kernel in footnote 1 on page 5, we admit that we did not stress it enough in the statement. We will reflect this comment by emphasizing the empirical validation provided by the paper mentioned above.

---

> > > > ### Comment · Reviewer_9inD · 2022-08-08
> > > > **Increase score**
> > > >
> > > > Dear authors,
> > > >
> > > > Thank you for your response. I think all of my concerns are addressed now. I decided to increase my score to 6. I can't increase more because of all the existing issues with the paper: the remarks/discussions around the theoretical results, and the clarity of the proof. The authors need to fix these issues in the final version of the paper.

---

> > > > > ### Author Response · Authors · 2022-08-08
> > > > > **Thanks for actively engaging in the discussion**
> > > > >
> > > > > We are happy to finally be able to resolve the main concerns. We will reflect the feedback/the discussion communicated in the revised version.

---

> ### Author Response · Authors · 2022-08-02
> **Response to the preliminary review from Reviewer 9inD - Others**
>
> ### Experiment
>
> We want to stress that the benefit of the method is more noticeable in the structure learning experiment where we use a much larger batch size and we compare with the baselines which may seem to perform similarly well on the benchmarks. This bigger benefit with the larger batch sizes is noteworthy considering the purpose of the batch acquisition trading compute resource with wall-clock run time.
> We conjecture that the performance degradation of LAW-EI is attributed to over-emphasis on the quality, i.e. more exploitation. Since its weight function is not bounded above, thus there is chance that the LAW-EI emphasizes the acquisition weights more, i.e. more exploitation in combination with EI’s emphasis on exploitation.
>
> In large search spaces like permutation spaces, $\beta$ in UCB becomes very large making UCB effectively consider the predictive uncertainty only. Therefore we conjecture that this will behave similarly to ordinary DPP which only considers the predictive variances. Moreover, in [Tarun Kathuria, Amit Deshpande, and Pushmeet Kohli. Batched gaussian process bandit optimization via determinantal point processes. Advances in Neural Information Processing Systems], UCB is marginally worse than EST in relatively small spaces (in terms of dimensionality in Euclidean cases), thus we concluded that LAW-UCB will not be able to balance between the quality and the diversity and excluded it. As shown in the structure learning experiments, there is a benefit of well-balanced consideration of the quality and the diversity, especially the batch size is large.
>
>
> ### Comparison to other non-permutation combinatorial BO on permutation problems
>
> We focus on batch acquisition. Since the choice of kernel and the choice of the batch acquisition is orthogonal, the comparison of combinations of other kernels and other batch acquisition methods is beyond the scope of this paper. Based on the previous results and our preliminary runs comparing different kernels, we picked the position kernel which is the best performer.
>
> In discrete&categorical BO, each variable is independent while in permutation BO, each variable is dependent. For example, for a given permutation $\pi = (p_1, \cdots, p_N)$, we have the constraints that $p_i \neq p_q$ for $i \neq j$. In discrete%categorical BO, the choice of one variable does not affect the choice of other variables. This additional constraint requires additional adjustment of the default settings of all discrete&categorical BOs. Also, our focus is on the comparison of the batch acquisition not on the comparison of different kernels, we focus on the batch acquisition comparison with the best permutation kernel we could find.

---

### Official Review · Reviewer_SzBU · 2022-07-20

**Rating:** 7
**Confidence:** 3
**Soundness:** 3 good
**Presentation:** 3 good
**Contribution:** 3 good

**Summary:**

This paper proposes a new method of batch Bayesian optimization for combinatorial optimization problems, especially permutation problems. The authors found that the density function of L-ensemble, one definition of a determinantal point process (DPP), can be used as a measure of batch point diversity to ensure the diversity of batch points acquired in each round of Bayesian optimization.

In order to guarantee the diversity of batch points while taking advantage of the proposal using the ordinary acquisition function, the authors introduced the Acquisition Weight Kernel (AWK) and proposed to acquire batch points by maximizing the density of L-ensemble induced by the AWK. Here, as an actual optimization method, since the determinant of the AWK matrix, which is the numerator of the L-ensemble density induced from AWK, can be written as the product of the determinant of the original kernel matrix and the monotonic transformation of the acquisition function, and since log det is submodular, a greedy algorithm is proposed to repeat the operation of maximizing the above product by one point at a time and then acquiring it batch size times.

In terms of theoretical contributions, it is first shown to achieve asymptotic riglet vanishing for UCB and EST strategies, similar to existing Bayesian optimization based on maximum information gain. The positive definiteness of the position kernel is rigorously proved, and an upper bound on the maximum information gain when using it as the kernel function is also derived.

In the experiments, the proposed method is compared with existing methods by three benchmark problems: the quadratic assignment problems (QAP), the flowshop scheduling problems (FSP), and the traveling salesman problems (TSP).

**Questions:**

- LAW-UCB has not been evaluated experimentally, but how well can it be expected to perform?

- Entropy search acquisition functions and ESTs seem similar, but can the former be used for search on permutation spaces (or especially can ES use in the proposed method)?

**Limitations:**

The authors adequately addressed the limitations and potential negative societal impact of their work.

**Strengths And Weaknesses:**

Strength

- For the BO problems on permutation space, the authors propose a batch point acquisition strategy that successfully extends the existing batch Bayesian optimization. The proposed method, LAW, can be used in combination with existing acquisition functions. In this paper, the combination with EST, EI, and UCB is considered, but it is also possible to combine with other acquisition functions.  indicating that it has high applicability. In addition, although the main focus of this paper is BO on permutation space, this batch acquisition strategy can also be used in batch Bayesian optimization on ordinary continuous space, suggesting its high applicability.

- Although the position kernel, the kernel for BO on the permutation space used in this study, has been proposed in existing studies, the authors proved its positive definiteness and show that using this kernel provides a theoretically tighter upper bound for the maximum information gain compared to the existing analysis.

- In the evaluation experiments with benchmark problems and structural learning problems, the proposed method showed good overall performance. The evaluation experiments also include an ablation study for the weignt function, which clearly demonstrates the importance of the introduction.

Weakness

- The setting of the weight function is arbitrary. While there are comparisons between cases without weights and with weights, there is a lack of discussion on how performance varies depending on how the weights are set.

- The EST on which the proposed method is based can be interpreted as evaluating the conditional probability that the candidate point x is the optimal solution, but the relationship with entropy search, which is one of the main acquisition functions with similar ideas, is unclear.

---

> ### Author Response · Authors · 2022-08-02
> **Response to the preliminary review from Reviewer SzBU**
>
> We thank for the reviewer's constructive feedback to further clarify the paper. Below we provide answers to the questions raised.
>
> ### Relation to the entropy search / Applicability of entropy search family to the permutation space
>
> Thanks for sharing the insight on the relation between two acquisition functions. We agree that their philosophy has something in common.
> However, in the context of combinatorial Bayesian optimization, the acquisition functions relying on GP function sampling (Thompson sampling) or random features including entropy search has computational issues.
>
> Even though there is recent progress on the efficient GP function sampling on continuous spaces, to our knowledge, those methods critically require the closed-form expression of eigenfunctions of the given kernel and such closed-form basis functions are available only for a few kernels, e.g. RBF. When a new kernel is used, using acquisition functions utilizing GP function samples or random features poses another challenge of finding closed-form expression of eigenfunctions of a given kernel. Therefore, the applicability of such methods depends on whether the kernel has the closed form expression for eigenfunctions.
>
> On small search spaces where the eigendecomposition of the gram matrix on the entire space is affordable, the acquisition functions using GP function samples or random features can be considered. But this limits the applicability of our method to up to $S_7$ where $|S_7|=7!=5040$.
>
> Thanks for pointing this out, we will discuss this point in the revised version.
>
>
> ### Expected performance of LAW-UCB
>
> The reason we exclude LAW-UCB is that UCB has a very large $\beta_t$ (UCB hyperparameter) due to the excessive size of permutation spaces and practically ignores the predictive mean. Essentially, LAW-UCB will become a predictive standard deviation weighted DPP thus no consideration of the quality and sole emphasis on the predictive uncertainty. Thus we expect LAW-UCB on such large search spaces at best performs similarly to DPP-MAX-EST which was shown to perform slightly better than DPP-MAX-UCB on continuous search space [Tarun Kathuria, Amit Deshpande, and Pushmeet Kohli. Batched gaussian process bandit optimization via determinantal point processes. Advances in Neural Information Processing Systems].
>
> In our experiment, especially in the structure learning problem, we observe the benefit of considering both the quality and the diversity simultaneously. We conjecture that such over-emphasis on diversity is likely to degrade the performance of the batch acquisition.
>
> ### How to set the weight function
>
> Please refer to our answer at (https://openreview.net/forum?id=LODRFJr96v&noteId=T604v0RpKE) to the same question

---

> > ### Comment · Reviewer_SzBU · 2022-08-09
> > **Thank you for the careful responses**
> >
> > Thank you for your detailed answer to my questions.
> > I was convinced that it is difficult to use existing information-theoretic acquisition functions on combinatorial spaces. I think this is a material to support the strength of the proposed method.
> > It is also understandable that UCB would actually do the exploration looking almost exclusively at the predicted variance. However, the practical performance expected from the combination of the proposed method and other acquisition functions suggests that the difficulty of practical application of UCB is not a problem.
> > From the above, I continue to recommend the acceptance.

---

### Comment · Area_Chair_jt37 · 2022-08-03
**Discussion period**

Thanks to all reviewers and authors for their work on this submission.

As the discussion period starts, I want to make sure that reviewers have read the author's response, and if needed react to it.

This can be done either by communicating with authors or in private conversation within the reviewing team.

Reviewer 9inD: Have your concerns been properly adressed by the authors?

---

> ### Comment · Reviewer_9inD · 2022-08-06
> **After reading the authors' response**
>
> Dear AC, the authors, and other reviewers,
>
> I've read the authors' response, some of my concerns have been addressed, but some still remain. Please see my detailed response under my original review.

---

> > ### Comment · Reviewer_9inD · 2022-08-08
> > **My main concerns are addressed**
> >
> > Dear AC, the authors, and other reviewers,
> >
> > I feel much better now with the paper, my main concerns have been addressed. I decided to increase the score to 6. More details can be found under my original review.

---

### Meta-Review · Area_Chair_jt37 · 2022-08-26

**Recommendation:** Accept
**Confidence:** Certain

**Metareview:**

This work introduces a new method, based on Bayesian optimization, for solving combinatorial problems. It uses determinantal point processes in their LAW method, which is applied to permutation problems.

The authors provide a theoretical analysis of their method, as well as empirical evidence on several of these problems.

The authors appropriately addressed the comments that the reviewers had, and they all feel comfortable to recommend to accept, and so do I.

**Award:**

No

---

### Decision · Program_Chairs · 2022-09-14

Accept